# Updated Neoadjuvant Treatment Landscape for Early Triple Negative Breast Cancer: Immunotherapy, Potential Predictive Biomarkers, and Novel Agents

**DOI:** 10.3390/cancers14174064

**Published:** 2022-08-23

**Authors:** Giovanna Garufi, Luisa Carbognin, Francesco Schettini, Elia Seguí, Alba Di Leone, Antonio Franco, Ida Paris, Giovanni Scambia, Giampaolo Tortora, Alessandra Fabi

**Affiliations:** 1Oncologia Medica, Fondazione Policlinico Universitario Agostino Gemelli IRCCS, 00168 Roma, Italy; 2Oncologia Medica, Università Cattolica Del Sacro Cuore, 00168 Roma, Italy; 3Department of Woman and Child Health and Public Health, Division of Gynecologic Oncology, Fondazione Policlinico Universitario Agostino Gemelli IRCCS, 00168 Roma, Italy; 4Medical Oncology Department, Hospital Clinic of Barcelona, 08036 Barcelona, Spain; 5Translational Genomics and Targeted Therapies in Solid Tumors, August Pi i Sunyer Biomedical Research Institute, 08036 Barcelona, Spain; 6Faculty of Medicine, University of Barcelona, 08036 Barcelona, Spain; 7Breast Unit, Department of Woman and Child Health and Public Health, Fondazione Policlinico Universitario Agostino Gemelli IRCCS, 00168 Roma, Italy; 8Unit of Precision Medicine in Senology, Department of Woman and Child Health and Public Health, Scientific Directorate, Fondazione Policlinico Universitario Agostino Gemelli, IRCCS, 00168 Roma, Italy

**Keywords:** triple-negative breast cancer, neoadjuvant chemotherapy, immune checkpoint inhibitors, platinum agents, PARP-inhibitors, target therapies, predictive biomarkers

## Abstract

**Simple Summary:**

In recent years, several agents have been tested in randomized clinical trials in addition to anthracycline and taxane-based neoadjuvant chemotherapy (NACT) in early-stage triple-negative breast cancer (TNBC) to improve pathological complete response rate and, ultimately, survival outcome. Platinum agents, immune checkpoint inhibitors (ICIs), and PARP-inhibitors are the most extensively studied, while established predictors of their efficacy are lacking. Based on the biological features of TNBC, the purpose of this review is to provide an overview of the role of platinum agents, immunotherapy, and novel target therapies in the neoadjuvant setting. Moreover, based on safety issues and financial costs, we provide an overview of potential biomarkers associated with increased likelihood of benefit from the addition of platinum, ICIs, and novel target therapies to NACT.

**Abstract:**

Triple-negative breast cancer (TNBC) is characterized by the absence of hormone receptor and HER2 expression, and therefore a lack of therapeutic targets. Anthracyclines and taxane-based neoadjuvant chemotherapy have historically been the cornerstone of treatment of early TNBC. However, genomic and transcriptomic analyses have suggested that TNBCs include various subtypes, characterized by peculiar genomic drivers and potential therapeutic targets. Therefore, several efforts have been made to expand the therapeutic landscape of early TNBC, leading to the introduction of platinum and immunomodulatory agents into the neoadjuvant setting. This review provides a comprehensive overview of the currently available evidence regarding platinum agents and immune-checkpoint-inhibitors for the neoadjuvant treatment of TNBC, as well as the novel target therapies that are currently being evaluated in this setting. Taking into account the economic issues and the side effects of the expanding therapeutic options, we focus on the potential efficacy biomarkers of the emerging therapies, in order to select the best therapeutic strategy for each specific patient.

## 1. Introduction

Breast cancer (BC) is the most commonly diagnosed tumor among females, with a life-time probability of developing invasive BC from birth to death of 12.8% [1]. Triple-negative breast cancer (TNBC), accounting for approximately 15 to 20% of all BCs, represents a heterogeneous group lacking the expression of estrogen and progesterone receptors (ER and PR), as well as human epidermal growth factor receptor 2 (HER2) amplification [2,3]. Epidemiological data show that TNBC mostly occurs in premenopausal young women under 40 years old and patients harboring a breast cancer susceptibility gene 1 or 2 (*BRCA1/2*) mutation [4,5], presenting at diagnosis with a larger tumor size, higher grading, and more frequent lymph node spread than other BC subtypes [6,7]. Triple negative BCs are characterized by a more aggressive biological behavior, with high early recurrence rate and worse survival when compared with the Luminal and HER2 positive BC [8,9]. Indeed, from a molecular perspective, BC is composed of at least four main so-called intrinsic subtypes, namely luminal A, luminal B, HER2-enriched and basal-like (BL), with different clinical behavior, prognosis, and response to different treatments [10]. Triple-negative BCs have been found to be mostly basal-like (BL), although not exclusively [11]. In fact, as showed by genomic investigations, a small fraction also consists of luminal A, luminal B, and HER2-enriched tumors, with currently unclear therapeutic implications [12]. Furthermore, microarray-based expression studies revealed six intrinsic subgroups of TNBC [12,13], each displaying specific molecular features and oncogenic signaling pathways. These subtypes encompass two BL (BL1 and BL2), a mesenchymal (M), a mesenchymal stem-like (MSL), an immunomodulatory (IM), and a luminal androgen receptor (LAR) type [14]. The BL1 subtype is enriched in cell cycle-regulating and DNA repair-related genes, while the BL2 subtype is characterized by myoepithelial markers and growth factor expression. The M and the MSL subtypes share expression of epithelial–mesenchymal transition and growth factor pathways, but a limited expression of genes involved in proliferation characterizes the MSL type. Finally, the IM subtype shows expression of immune signal pathways, while the LAR subtype is enriched in luminal gene and androgen receptor expression [15]. However, the initial six subgroups were later redefined to four, since evidence was provided that the IM and MSL TNBC subtypes consist of tumors with a substantial component of tumor-associated cells, i.e., immune and stromal cells, respectively [12]. 

The underlying profound biological diversity of what was traditionally thought of as a single tumor entity likely explains the lack of appropriate actionable targets [16]. Therefore, chemotherapy has traditionally been the mainstay of TNBC treatment until recently [11,17]. Anthracyclines and taxanes have historically been considered the most active drugs; however, platinum agents have been added in the metastatic and neoadjuvant settings with encouraging results [18,19,20].

However, the efficacy of neoadjuvant chemotherapy remains poor, and residual disease will eventually lead to tumor recurrence. Therefore, it is urgent to develop new treatment strategies and targets. Given the constant advances in TNBC, the purpose of this review is to provide an update of new chemotherapy regimens in the neoadjuvant setting of TNBC, the role of immunotherapy in early TNBC and its potential biomarkers of response, and the role of novel biologic and targeted agents.

## 2. Pathological Response after Different NACT Regimens and Potential Predictive Role of *BRCA* and HRD Status

Neoadjuvant chemotherapy represents the standard of care for locally advanced or inoperable BCs, but it is also used for operable BCs when conservative surgery is not feasible [21]. The pathological complete response (pCR), defined as the absence of residual invasive disease in both breast and regional lymph nodes after NACT, predicts long-term outcomes and is therefore a potential surrogate marker for survival and a reliable endpoint for neoadjuvant clinical trials [22,23]. 

In recent years, a large amount of literature has shown that the use of NACT regimens in the treatment of TNBC has a significantly higher pCR than for hormone receptor-positive breast cancer, and can ameliorate the prognosis of TNBC patients [24,25]. Thus, selection of appropriate chemotherapy drugs and optimization of chemotherapy regimens are crucial for guaranteeing good treatment outcomes and prognosis of TNBC patients. The standard anthracycline and taxane-based chemotherapy regimens yield a pCR rate for about 30–40% of TNBC patients, with an estimated 10-year relapse-free survival of 86% for patients who achieve pCR vs. 23% for those with significant residual disease after chemotherapy [26]. Efforts to further enhance pCR rates are under way by assessing the addition of different chemotherapeutic agents and novel biologic and targeted agents to standard regimens. 

### 2.1. Incorporation of Platinum Agents

The addition of platinum-based chemotherapy has been proposed. Carboplatin, with its alkylating cytotoxic activity, is a DNA damaging agent that exerts a synergistic effect when added to a taxane-based NACT, as demonstrated by clinical as well as preclinical studies [27]. Three randomized trials have shown that carboplatin, when added to an anthracycline and taxane-based NACT, results in an increase in pCR rate between 10% and 15%, although at the cost of greater hematological toxicity, but with promising results on survival outcomes [28]. 

Firstly, the CALGB 40603 trial showed that the addition of carboplatin to dose dense AC and T in stage II and III TNBC (n = 443) significantly increased pCR rates in both the breast (60% vs. 44%; *p* = 0.0018) and axilla (54% vs. 41%; *p* = 0.0029). The study was not powered to assess EFS or OS, and no significant differences were found in these measures, although a trend in EFS was seen [19]. 

In the other study, GeparSixto, 66 randomized stage II and III TNBC patients (n = 595) received carboplatin weekly with a backbone chemotherapy regimen consisting of weekly paclitaxel, liposomal doxorubicin, and bevacizumab for 18 weeks. Similar to what was observed in the CALGB trial, an improvement in pCR rates was found (37% to 53% with carboplatin; *p* = 0.005), with also an improvement in DFS in TNBC patients randomized to the carboplatin arm (85.8% vs. 76.1%, HR = 0.56; *p* = 0.035) [18]. 

More recently, results from the BrighTNess trial, a randomized phase III clinical trial (n = 634) of three treatment arms (paclitaxel alone (P), paclitaxel and carboplatin (PCv), and paclitaxel, carboplatin, and veliparib (PCbV)) have been updated. pCR significantly improved when adding carboplatin (58% vs. 31%), while further addition of veliparib did not lead to improvements (53%). In the same direction, the addition of carboplatin improved the 4-year EFS from 69% to 79% (HR 0.57, 95% CI 0.36–0.91, *p* = 0.018), while the further addition of veliparib did not (HR 0.63, 95% CI 0.43–0.92, *p* = 0.016). Overall survival data are still immature, but head to the same direction [29].

Importantly, a very recent trial-level and patient-level meta-analysis further confirmed that the addition of carboplatin to standard anthracycline/taxane-based (neo)adjuvant chemotherapy was able to improve pCR rates, but also showed for the first time a clear benefit in terms of DFS and OS [30].

Overall, the evidence published so far support the use of carboplatin in addition to standard NACT, at least in high-risk patients. Clinical trials are ongoing to further evaluate the role of platinum agents in patients with both *BRCA* mutated (*BRCA*m) and *BRCA* wild type TNBCs.

The addition of other agents to standard anthracycline and taxane-based chemotherapy regimens, including gemcitabine, capecitabine, and bevacizumab, did not improve outcomes in TNBC [19,31,32].

### 2.2. De-Escalation Strategies

In addition to the pCR improvements with chemotherapy agents that are added to the standard anthracyclines and taxane-based NACT, attempts to de-escalate neoadjuvant therapy without anthracyclines have also been made. 

Carboplatin plus taxane regimens have demonstrated an encouraging pCR rate with a favorable toxicity profile in TNBC patients [33,34,35]. In this regard, Sharma et al. conducted NeoSTOP, a neoadjuvant phase II trial, in which 100 patients with stage I–III TNBC were randomized to a standard anthracycline and taxane-based regimen, or to carboplatin and docetaxel. Both arms achieved comparable pCR rates (54%), EFS and OS, but with a much more favorable toxicity profile and lower health care costs for the anthracycline-free regimen [36]. Thus, non-anthracycline regimens may be an appropriate option for some patients, such as older patients and those with considerable comorbidities and cardiac risks.

### 2.3. Predictive Role of BRCA Mutations and HRD

In recent years, the predictive role of *BRCA1/2* germline mutations and homologous recombination deficiency (HRD) has been explored. Subgroup analyses of randomized trials have shown that *BRCA1/2* mutated tumors have higher pathologic response rates than wild-type tumors [18,28,29]. Nonetheless, contrary to expectations, patients with a germline *BRCA* wild type exhibited a larger increase in pCR rates than *BRCA*m patients with the addition of carboplatin to NACT [28,37]. 

The HRD score is the quantitative result of genomic instability resulting from epigenetic inactivation of *BRCA*, mutations in other genes, or post-translational modifications of other key proteins involved in the homologous recombination system [38]. A threshold of HRD score ≥ 42 defines a BRCAness phenotype, identifying sporadic TNBCs with DNA repair defects and clinical and treatment response features similar to *BRCA* mutated patients [38,39]. Based on the data from available retrospective studies, the effect of platinum addition on pCR rate seems globally more pronounced in TNBCs with high HRD compared to those with a low HRD score [40]. Therefore, the presence of a pathogenic variant of *BRCA1/2* genes could represent a more robust genomic scar than the genomic instability of the BRCAness phenotype, which could instead identify tumors that benefit from the addition of an alkylating agent such as platinum to standard NACT [41]. Further investigations could be important to define the inclusion of the HRD as a valid marker for therapeutic choice as an alternative to the *BRCA* mutation.

## 3. Neoadjuvant Immunotherapy

### 3.1. Biological Rationale

Tumor cells can evade recognition and destruction by the host immune system. To date, many immune escape mechanisms have been identified, including expression of the immune checkpoint system that normally terminate immune responses after antigen activation [42]. The activation of the immune checkpoints, such as PD-1 and PD-L1, results in suppression of the infiltrating T-cells functions [43]. Thus, blocking the immune checkpoint system is a promising treatment strategy for accomplishing effective antitumor immunity [44].

Breast cancer has historically been considered immunologically quiescent. Nonetheless, immune checkpoint inhibitors (ICI) have been successfully tested even in this cancer type, especially in the TN subgroup. In fact, it is known that triple-negative tumors are enriched in immune cells, in particular T lymphocytes and myeloid cells, when compared to luminal cancers. Of note, Wagner et al. demonstrated that a higher number of immunosuppressive cells, such as T regulator lymphocytes, PD-1-positive lymphocytes, and PD-L1-positive myeloid cells, are found in TNBCs compared to luminal tumors [45]. Furthermore, TNBC is characterized by a relatively high tumor mutational burden (TMB) compared to other subtypes of BC [46]. These properties confer TNBC patients as good candidates for immunotherapy with PD-1 and PD-L1 inhibitors [47]. As a matter of fact, immune checkpoint blockade with pembrolizumab and atezolizumab (now withdrawn) has recently been approved for advanced-stage PD-L1 positive TNBC based on the improvement in outcomes seen in KEYNOTE-355 and IMpassion130 when combined with frontline chemotherapy [48,49,50,51,52].

On the other hand, it is known that NACT is able to reduce the count of immunosuppressive cells and induce immunogenic cell death in the host, reactivating the anti-tumor immune response [53,54]. Furthermore, a study of genomic and transcriptomic profiling of primary and metastatic TNBC samples revealed a shift towards a BL genotype and a reduction of the IM subtype, with concordantly reduced immune-activated gene signatures and tumor infiltrating lymphocytes (TILs), in recurrent TNBCs compared to paired primary tumors [55]. Thus, there is a strong rationale for ICI administration early in the disease course. 

### 3.2. The Current Neoadjuvant Landscape of Randomized Immunotherapy Trials

Several studies have investigated the impact of the addition of immunotherapy agents to NACT in TNBC. To date, results have been published from phase I, II, and III studies evaluating the addition of pembrolizumab, atezolizumab, or durvalumab, to different chemotherapy backbones. KEYNOTE-173 and I-SPY2 were the first two multicohort studies, phase I and II, respectively, that evaluated the addition of pembrolizumab to anthracycline and taxane-based NACT, showing a pCR rate of 60% [56,57]. In particular, the I-SPY2 trial demonstrated a 40% increase in estimated pCR rate (from 22% in the control arm to 60% in the experimental arm) [57]. Based on these promising first results, different randomized studies were conducted, as summarized in Table 1. 

The GeparNuevo phase II trial evaluated the addition of durvalumab to a standard anthracycline and taxane-based NACT in 174 early TNBC patients [58]. One third of randomized patients (32%) had node-positive disease and 87% were PD-L1 positive. Patients received durvalumab/placebo monotherapy two weeks prior to the start of chemotherapy (window-phase), followed by durvalumab/placebo every 4 weeks for three cycles plus nab-paclitaxel (weekly for 12 weeks), and then by durvalumab/placebo every 4 weeks for two cycles plus dose-dense (DD) epirubicin and cyclophosphamide (EC) for four cycles. The primary endpoint was the pCR rate (ypT0 ypN0). The study was amended, and the window phase was interrupted after 117 patients were recruited. Subsequently, all patients started with durvalumab/placebo plus chemotherapy on the first day. While the addition of durvalumab numerically increased pCR rates in all patients, this increase did not reach statistical significance (53.4% vs. 44.2%; OR 1.45, 95% CI 0.80–2.63; *p* = 0.224). Of note, among the 117 patients who participated in the “window phase”, receiving a single dose of their assigned study group prior to NACT, those who received durvalumab had a significantly higher pCR (61.0% vs. 41.4%, OR 2.22, 95% CI 1.06–4.64; *p* = 0.035), suggesting a potential benefit for this kind of “immune priming”. At a median follow-up of 42.2 months, survival results showed an improvement of 3-year IDFS from 76.9% to 84.9% with the addition of durvalumab (HR 0.54, *p* = 0.0559), irrespective of whether patients received the window phase or not. Subgroup analysis demonstrated the benefit from durvalumab in both pCR patients and non-pCR patients, with a 3-year IDFS improvement from 86.1% to 95.5% and 69.7% to 76.3%, respectively [59]. 

In the NeoTRIPaPDL1 phase III study, 280 patients with early TNBC were randomized to receive carboplatin and nab-paclitaxel on days 1 and 8 with or without atezolizumab on day 1 every 3 weeks for eight cycles. The neoadjuvant treatment was followed by surgery and then by anthracycline regimen for four cycles as per investigator choice. The primary aim of the study was to compare event-free survival (EFS) at 5 years, while the pCR rate (ypT0/Tis ypN0) was a secondary endpoint. Most of the patients (87%) had node-positive disease, and 56% had a PD-L1-positive tumor. The pCR rate was 43.5% with atezolizumab and 40.8% without atezolizumab (*p* = 0.066) [60]. Continuing follow-up for the EFS is ongoing. 

The KEYNOTE-522 trial was the first prospective randomized controlled phase III study to show a benefit from adding pembrolizumab to NACT in early TNBC patients. The trial randomized 1174 patients to receive pembrolizumab or placebo every 3 weeks for four cycles with paclitaxel (weekly for 12 weeks) and carboplatin (weekly for 12 weeks or every 3 weeks for four cycles), followed by pembrolizumab or placebo every 3 weeks for four cycles plus doxorubicin and cyclophosphamide (AC) or EC every 3 weeks for four cycles. After surgery, patients received adjuvant pembrolizumab or placebo every 3 weeks for up to nine cycles. Co-primary endpoints were the pCR rate (ypT0/Tis ypN0) and EFS in the ITT population. Half of the patients (51%) had a node-positive disease, and the majority (82%) had a PD-L1-positive tumor. In this trial, the percentage of patients who responded to treatment was significantly higher in the pembrolizumab arm (64.8% vs. 51.2%, delta 13.6, *p* = 0.0005). The efficacy of pembrolizumab was shown to be consistent regardless of PD-L1 status [61]. Moreover, updated survival results were recently made available, showing, at a median follow-up of 39 months, a clinically and statistically significant benefit with pembrolizumab addition to NACT with an improvement of the 3-year EFS from 76.8% to 84.5% (HR 0.63, *p* = 0.0003) [61]. Interestingly, exploratory analysis showed that the greatest relative benefit from pembrolizumab addition derives from patients at higher risk of relapse. The 3-year EFS rate improved from 56.8% to 67.4% and from 92.5% to 94.4% in the invasive residual disease and pCR subgroups, respectively [62]. Based on these data, in July 2021, pembrolizumab received FDA approval for early-stage high-risk TNBC in combination with chemotherapy as neoadjuvant treatment, and then continued as a single agent as adjuvant treatment after surgery.

The IMpassion031 study was the last published study evaluating the addition of an ICI to NACT. In this randomized phase III trial, 333 patients received atezolizumab or placebo every 2 weeks for six cycles in combination with nab-paclitaxel (weekly for 12 weeks), followed by atezolizumab or placebo every 2 weeks for four cycles with DD AC for four cycles. After surgery, patients in the atezolizumab arm continued to receive immunotherapy every 3 weeks for 11 cycles. The co-primary endpoints were pCR rate (ypT0/Tis ypN0) in both the entire and the PD-L1-positive populations. EFS in both populations was included among the secondary endpoints. Results revealed that 46% of patients had a PD-L1-positive tumor, and 34% of patients in the experimental arm and 43% in the control arm had a node-positive disease. Significant improved pCR was found in the atezolizumab arm (58% vs. 41%, *p* = 0.004), regardless of PD-L1 status, disease stage, or lymph node status. Regarding survival, a favorable trend was found for the experimental arm; however, with a median follow-up of 20.6 months, the EFS outcomes were still immature (HR 0.76; 95% CI, 0.40 to 1.44) [63]. 

### 3.3. Potential Predictive Biomarkers of Neoadjuvant Immunotherapy Benefit

The addition of ICI to neoadjuvant chemotherapy has been shown to significantly improve pCR and event-free survival for stage II–III TNBC, thus settling down as a new standard of care in this setting. This achievement, however, has raised several questions. Identification of reproducible biomarkers to predict who may or may not benefit from ICI addition upon risk or relapse and immunological background is currently a major challenge in this field.

#### 3.3.1. Tumor-Infiltrating Lymphocytes and Programmed Cell Death-Ligand 1

Tumor-infiltrating lymphocytes (TILs) and programmed cell death-ligand 1 (PD-L1) expression have been the most explored biomarkers in the neoadjuvant setting. Clinical studies have well established the prognostic and predictive value of TILs in the BC neoadjuvant setting [64,65]. Tumor infiltrating lymphocytes are defined by the percentage of area of the stromal tissue covered by mononuclear cells (lymphocytes and plasma cells). Initially, stromal (sTILs) and intratumoral lymphocytes (iTILs) were evaluated separately. Nowadays, however, since TILS moves within the tumor microenvironment (TME) continuously, this distinction is considered artificial and related to the static histological samples [66]. Both sTILs and iTILs have been found to be predictive of pCR to NACT [64]. 

In the GeparNuevo trial, patients were stratified by sTILs levels (low 0–10% vs. intermediate 11–59% vs. high ≥ 60%). Twenty-five patients (14.4%) had high sTILs, 47.7% had intermediate sTILs, while 37.9% had low values. Stromal TILs, as a continuous variable, were significantly associated with pCR in the entire cohort; however, they were not specific for durvalumab response. Interestingly, during the window-phase, iTILs increased in both arms. Change of iTILs between baseline and after the window-phase predicted pCR with durvalumab, but the interaction test for iTILs change and therapy arm did not formally meet a statistical significance, likely due to the small sample size. Regarding PD-L1 expression, an improved response in PD-L1-positive tumors was observed, regardless of treatment arm, with an absolute pCR difference rate greater for the PD-L1-positive than for the negative population. However, in the durvalumab arm, the pCR rate did not significantly differ between PD-L1-positive tumors and the PD-L1-negative group (58.0% vs. 44.4%, respectively; *p* = 0.445). Consequently, in the GeparNuevo trial, an increased expression of both TILs and PD-L1 were associated with an improvement of pCR rate to NACT without specifically predicting benefit for durvalumab response [58]. The correlation between baseline TILs and PD-L1 was also assessed in the NeoTRIPaPDL1 study. Of note, the value of both sTILs and iTILs was higher in the control group compared with the experimental group, possibly masking pCR differences between treatment arms. Both TILs and PD-L1 were significantly correlated with pCR in the atezolizumab group, while only PD-L1 expression was associated with pCR in the placebo group. Again, there was a trend towards a better pathological response to NACT among PD-L1-enriched tumors but, contrary to the results of the GeparNuevo, the relationship between TILs and pCR was only significant for the atezolizumab arm. Interestingly, most patients had a robust increase of TILs after one cycle of treatment, mostly driven by chemotherapy, since it was similar among both arms. Notably, TILs change was predictive of pCR and, in the atezolizumab arm, tumors with sTILs ≥ 40% had a significantly higher probability of pCR than those who had sTILs < 40% [67]. Once more, however, TILs and PD-L1 expression seemed to predict benefit in both arms, thus questioning their use to select patients for immunotherapy.

In the KEYNOTE-522 study, both PD-L1-negative and PD-L1-positive patients derived the same benefit from pembrolizumab addition. However, pCR rates were also higher in PD-L1-enriched tumors, regardless of treatment arm (68.9% vs. 54.9% in the PD-L1-positive patients, whereas 45.3% vs. 30.3% the PD-L1-negative patients). However, it is important to mention that the PD-L1 threshold used in the subgroup analyses (CPS ≥ 1) may not be the optimal one, since a higher threshold (CPS > 10) is currently being used for patients’ selection in the metastatic setting [68]. The assessment of the association between pCR and TILs count is expected [61]. 

A similar absolute difference in the pCR rate on the basis of PD-L1 status was also found in the IMpassion031 trial (69% vs. 49% in the PD-L1-positive tumors, whereas 48% vs. 34% in the PD-L1-negative tumors) [63]. Again, PD-L1 expression correlates with the pCR rate but does not predict immunotherapy benefit. No data on the correlation between TILs and pCR rate in the IMpassion031 were reported. Table 2 summarizes the pCR rate according to PD-L1 status (positive vs. negative) and the treatment arms, as reported in randomized trials.

Finally, the 14-gene IGG immune signature [69], tracking distinct biologic information compared to TILs, was recently found to be highly prognostic in early TNBC, beyond clinicopathological status [70]. A risk score integrating tumor size, nodal staging, and the IGG signature might be clinically valuable to help de-escalate systemic therapy in TNBC. 

#### 3.3.2. Tumor Mutational Burden and Microsatellite Instability

Tumor mutational burden (TMB) has also been evaluated as a potential predictor of response to ICI therapy. This biomarker is a measurement of the total number of non-synonymous mutations per coding area of the tumor genome [71]. A higher TMB is associated with the development of neoantigens, triggering recognition by the immune system and therefore T-cell proliferation [72]. In BC, few studies have evaluated the correlation of TMB and response to ICI. Hypermutated tumors, defined as ≥ 10 mutations per megabase (Mut/Mb), are found only in approximately 5% of BCs and are considerably enriched in TNBC compared to luminal cancers and in metastatic vs. primary tumors [71]. In the metastatic setting, the phase II TAPUR study showed that in 28 heavily pretreated advanced TNBC patients with high TMB, pembrolizumab treatment achieved remarkable activity with an objective response rate of 21% and a median PFS of 10 weeks [73]. 

In the neoadjuvant setting, a subgroup analysis of the GeparNuevo study also explored the role of TMB in predicting treatment response, founding a median TMB of 1.52 Mut/Mb. Of interest, continuous TMB predicted pCR in a multivariate analysis (*p* = 0.0012), regardless of durvalumab addition to NACT [74]. After TMB dichotomization at the upper tertile (2.05 mut/Mb), in both experimental and control arms, pCR rates were improved for high TMB compared with low TMB: 63% vs. 40%, respectively, in the durvalumab arm (*p* = 0.028) and 52% vs. 37%, respectively, in the placebo arm (*p* = 0.232). No interaction was observed between TMB and the treatment arm.

Tumors with dMMR and MSI-H have also been recognized to be sensitive to ICIs, probably due to the neoantigen generation [75]. Indeed, in 2017, the FDA approved pembrolizumab for any unresectable or metastatic solid MSI-H/dMMR tumors [76]. Concerning BC, only 1% of the tumors were found to be MSI-H. Of note, MSI-H BCs mostly overlap with those that are TMB-high [77]. However, none of the randomized trials have so far explored microsatellite instability (MSI) or mismatch repair deficiency (dMMR) as a predictor of response to neoadjuvant treatment with ICI.

#### 3.3.3. Novel Biomarkers

Other novel promising biomarkers are emerging, such as the detection of circulating tumor DNA (ctDNA), gains in *CD274* gene (which encodes for PD-L1), expression of major histocompatibility complex (MHC)-II on tumor cells, and gene-expression profiles (GEPs). In fact, the predictive value of gene-expression profiles has been recently studied in the NeoTRIPaPDL1 trial. RNA-sequencing was performed pre-treatment (n = 242) and pre-cycle 2 (n = 161). Strikingly, pre-treatment binary IO score predicted pCR in the atezolizumab arm (OR 3.64, 95% CI 1.68–7.90; *p* = 0.001), but not in the placebo one (OR 1.31, 95% CI 0.64–2.67; *p* = 0.046). Furthermore, high responders in the atezolizumab arm had high expression of some immune signatures. In the opposite way, high angiogenesis and fatty acid/cholesterol were independently correlated to resistance only in the atezolizumab arm (*p* = 0.005) [78]. 

#### 3.3.4. Lymph Node Status as A Predictive Factor of Immunotherapy Response

Based on the importance of the lymph node status in defining the risk of recurrence but also in the potential sensitivity to ICIs [79,80,81], it is also interesting to explore the effect of ICI according to nodal burden. In the above-mentioned randomized studies, pCR rate was reported according to the disease stage. Two studies (KEYNOTE-522 and IMpassion031) reported the pCR rate according to node status (positive vs. negative) [61,63], while the GeparNuevo and NeoTRIPaPDL1 trials distinguished the pCR according to disease stage (≥ IIA vs. < IIA and ≥ IIB vs. < IIB, respectively) [58,60]. 

In the NeoTRIPaPDL1 trial, where most patients had positive lymph nodes, no substantial differences in pCR rate were observed depending on disease stage (≥ IIB vs. < IIB) [60], whereas in the GeparNuevo, a trend towards a higher pCR rate was observed with the addition of durvalumab in patients with a more advanced stage [58]. In the KEYNOTE-522 study, node positive patients appeared to derive greater benefit in terms of pCR from pembrolizumab addition: pCR increased from 44.1% to 64.8% in the node-positive population (delta 20.7%) vs. from 58.6% to 64.9% (delta 6.3%) in the node-negative population [61]. However, at the survival analysis, no difference in terms of EFS benefit was observed, depending on nodal status [62]. Similarly, in the IMpassion031 trial, the difference rate of pCR after atezolizumab addition was higher in the node-positive population (26%) vs. the node-negative population (9%) [63]. 

Thus, node-positive patients appear to derive a greater benefit in pCR compared with node-negative patients. A plausible explanation can be found on the microenvironment of tumor-draining lymph nodes (TDNs). Indeed, it has been shown that T-cell response to checkpoint blockade relies not only on the reinvigoration of pre-existing dysfunctional TILs, but also on the expansion of new clonotypes that that may have recently entered the tumor [82]. Several studies report that TDNs have a prominent role in supplying T cell precursors that will become cytotoxic effector cells following checkpoint blockade [81,83].

Furthermore, recently published data indicate that TNBC patients with clinically enlarged lymph-nodes, defined as minimum diameter > 15 mm assessed by ultrasound before surgery, but with surgically resected, pathologically confirmed node-negative (pN0) disease have improved survival compared with pN0 and clinically small lymph nodes, defined as maximum diameter < 5 mm by ultrasound before surgery. This suggests that clinically enlarged but pathologically negative lymph nodes may serve as markers of early systemic anti-tumor immune response, conferring a survival advantage [79]. Based on these data, it would be interesting to know whether the identification of node-positive patients in the neoadjuvant above-mentioned trials was based exclusively on clinical assessment or also on node biopsy. Since inflamed but not metastatic lymph nodes also exhibit an immune-inhibited microenvironment, the addition of ICI to NACT could allow for improved pathological tumor response in both metastatic and enlarged, but not metastatic, lymph nodes. Distinction between metastatic and enlarged but pathologically negative lymph nodes by node biopsy at diagnosis could allow differential analysis of the benefit from the addition of ICI, explaining whether their benefit occurs only in the presence of metastatic lymph nodes, or also in the presence of inflamed lymph nodes, and therefore somewhat immunoinhibited, but not involved by the disease.

### 3.4. Conflicting Results: The Sensitizing Role of a Chemotherapy Backbone in Immunotherapy Response

It is not known whether the selection of a chemotherapy partner is critical to obtain the potential benefit from the addition of ICI. All the above-mentioned trials adopted a chemotherapy backbone based on anthracyclines and taxanes, except for the NeoTRIPaPDL1 trial. In this study, chemotherapy included a combination of carboplatin and nab-paclitaxel exclusively, without anthracyclines, which were administered at investigator discretion in the post-neoadjuvant setting [60]. 

Remarkably, the adaptive, non-comparative TONIC trial investigated strategies to enhance sensitivity to PD1-blockade in TNBC. In this phase II study, 67 pretreated TNBC patients received nivolumab without induction or with a two-week induction of one of the following treatments: radiotherapy, cyclophosphamide, cisplatin, or doxorubicin. The highest response rate (35%) was observed after doxorubicin. In this line, following doxorubicin induction, an upregulation of PD-1/PD-L1 and T-cell cytotoxicity pathways were detected in seriate biopsies with an increase in TILS and CD8 cytotoxic T cells, which were then even more pronounced after treatment with nivolumab [84]. These findings suggest that anthracyclines, more than other chemotherapeutic agents, may be able to modify the TME, turning ‘cold’ into ‘hot’ tumors and priming cancer cells for ICIs response. To support this hypothesis, the performance in pCR rate of the control arm of the above-mentioned studies appears to be concordant, while the pCR rate in the experimental arm of the NeoTRIPaPDL1 study falls below expected. 

In addition, KEYNOTE-522 is the only immunotherapy trial that has so far used carboplatin plus standard NACT with anthracyclines and taxanes, allowing the addition of pembrolizumab to the regimen associated with the highest pCR rate reported among TNBC patients. However, new research efforts are needed to clarify the need for platinum and anthracyclines when adding ICI, and to understand if more flexibility is admissible regarding the backbone chemotherapy. For greater clarity, results are expected from two trials: the randomized phase III GeparDouze (NCT03281954) trial, which is exploring the addition of atezolizumab to an anthracycline, carboplatin, and taxane-based NACT; and the single arm phase II NeoPACT (NCT03639948) trial, which is testing the addition of pembrolizumab to a carboplatin and docetaxel-based NACT regimen, without anthracyclines [85]. 

## 4. Novel Agents

### 4.1. PARP Inhibitors

PARP is a class of DNA repair enzymes. Its function is to maintain genome stability, repair DNA, and participate in cell progression and apoptosis. Inhibition of PARP will lead to the loss of DNA repair function, hence inducing apoptosis. PARP inhibitors have significant antitumor effects on *BRCA*-mutated tumors. *BRCA* mutations can be found in nearly 20% of TNBC patients, compared to 5% in other BC subtypes [86]. Promising results have been seen with the use of PARP inhibitors in monotherapy, such as olaparib and talazoparib, when compared to chemotherapy in germline *BRCA*-mutated (g*BRCA*m) metastatic breast cancer patients [87,88]. 

PARP inhibitors have also been studied in the neoadjuvant setting of TNBC patients. Talazoparib achieved encouraging pCR in patients with g*BRCA*m BC, including TNBC and hormone-receptor positive BC, as a neoadjuvant single-agent without the addition of chemotherapy [89]. A larger neoadjuvant Phase II trial (NCT03499353) is ongoing to further evaluate its role in this context.

The addition of PARP inhibitors to standard NACT is also under investigation. In the adaptive I-SPY2 trial, the addition of veliparib and carboplatin to standard NACT with anthracyclines and taxanes improved pCR from 26% to 51% [90]. To clarify the role of veliparib, the phase III BrighTNess trial was conducted, assessing the addition of veliparib plus carboplatin vs. carboplatin alone to standard NACT for high-risk stage II–III TNBC. Disappointingly, no benefit was observed with the addition of veliparib [91]. 

In the I-SPY2 trial, the combination of a PARP inhibitor plus an immune checkpoint inhibitor (olaparib plus durvalumab) added to neoadjuvant paclitaxel was also tested, improving pCR in TNBC from 27% to 47% [92]. Other clinical studies are ongoing to assess the possible synergy between PARP inhibition and ICI in early TNBC.

The clinical efficacy of PARP inhibitors for early stage TNBC irrespective of *BRCA* mutations, however, remains unclear, and their use is not routine in clinical practice. Encouraging results were observed in the OLTRE window of the opportunity trial [93], but results are awaited from larger ongoing trials (e.g., PARTNER trial–NCT03150576). 

### 4.2. Androgen Receptor (AR) Targeted Agents

Androgen receptor (AR) positivity has been reported in 10–15% of TNBC patients [94], and is a potential target for anti-androgen therapy. Patients with AR+ TNBC have a decreased chance of achieving pCR to neoadjuvant chemotherapy [17,95]. Promising responses have been reported with abiraterone, bicalutamide and enzalutamide in advanced AR+ TNBC patients [96,97,98]. Thus, clinical trials are ongoing also in early stage TNBC (e.g., NCT02689427).

### 4.3. Antibody–Drug Conjugates: Sacituzumab Govitecan

Sacituzumab govitecan is a novel antibody–drug conjugate composed of the anti-TROP2 component linked to the antineoplastic drug SN-38, the active metabolite of irinotecan. It has been recently approved following the results of the phase III ASCENT trial, which demonstrated significant improvement in both PFS and OS with sacituzumab govitecan when compared to single-agent chemotherapy for the treatment of relapsed or refractory TNBC [99]. In the early setting, it could provide an opportunity for de-escalation of traditional chemotherapy, and it is currently being evaluated in patients with residual disease after neoadjuvant treatment alone or in combination with atezolizumab in different ongoing trials (SASCIA trial–NCT04595565 and ASPRIA trial–NCT04434040).

## 5. Treatment of Residual Invasive Disease

The presence of a residual invasive disease after neoadjuvant treatment causes a risk of up to 30–40% of distant relapse, typically within the first three years after initial diagnosis [24,25]. Recently, the role of adding further adjuvant therapy in TNBC patients with significant residual disease after NACT has been evaluated. Response-guided therapy with differentiated treatment strategies for patients with or without residual disease after neoadjuvant treatment is key to optimize treatment of TNBC.

The CREATE-X study randomized patients with HER-2 negative BC and residual invasive disease after standard anthracycline and taxane-based NACT to adjuvant treatment with capecitabine for up to eight cycles. The TNBC subtype experienced a 42% reduction in risk of death with capecitabine [100]. 

The role of olaparib for patients with *BRCA* mutated BC with residual disease after neoadjuvant treatment have also recently been assessed in a large trial of 1836 patients. The 3-year EFS observed in the olaparib arm was significantly higher than in the placebo group (85.9% vs. 77.1%, delta 8.8%, *p* < 0.001) [101]. Based on these results, FDA has recently approved (March 2022) the use of adjuvant olaparib for *BRCA*-mutated Her2-negative high risk early BC patients [102].

To date, we have no information for patients with *BRCA* mutated BC with residual disease of whether pembrolizumab could be associated with olaparib or if a rationale subsists for a therapeutic sequence between the two agents; further investigations on this issue could be a challenge.

Finally, adjuvant-only immunotherapy could also have some clinical benefit on this defiant population. Results are awaited from trials testing the addition of ICI for patients not achieving pCR after neoadjuvant chemotherapy alone (aveluma in the A-BRAVE trial–NCT02926196 and pembrolizumab in the SWOG S1418/BR006–NCT02954874).

In the post-neoadjuvant setting, tools are being investigated to detect minimal residual disease (MRD) in the circulation in order to predict in which patients the disease will recur. An established method to detect MRD is the analysis of circulating tumor DNA (ctDNA), a subfraction of cell-free DNA present in blood plasma that results from apoptosis and necrosis of tumor cells [103,104]. In TNBC patients with residual disease at the time of surgery following NACT, ctDNA detection was significantly associated with lower distant disease-free survival, DFS and OS [105]. In addition, ctDNA detection is currently being studied to tailor ICI treatment escalation after completion of primary TNBC treatment. The randomized phase II c-TRAK-TN study (NCT03145961) consists of a serial ctDNA surveillance component and a therapeutic component with pembrolizumab for one year in case of ctDNA detection, in order to evaluate both the use of ctDNA to detect cancer cells after standard treatment and additional pembrolizumab in early-stage TNBC. 

## 6. Early TNBC and Chemoimmunotherapy: Implications for Pregnancy and Fertility

Breast cancer is the most commonly diagnosed cancer in pregnancy [106], with an increasing incidence in recent decades due to the tendency to delay pregnancy to a later age [107]. Pregnancy-associated breast BC (PABC), also known as gestational BC, is defined as BC diagnosed during pregnancy or up to the first year postpartum [108]. Compared with non-pregnant age-matched women, PABC is more frequently characterized by ta riple-negative immunophenotype [109] and an enriched mismatch repair deficiency mutational signature [110]. Patients with PABC have a poorer clinical outcome with higher mortality than nulliparous women [111]. Possible reasons lie in both the higher incidence of TNBC and the more advanced disease stage at diagnosis, due to the delay in diagnosis related to physiological gestational changes [112]. Local staging involves breast ultrasound, including the axillary area [113], while mammography may confer a minimal dose to the fetus [114], and the safety of gadolinium of contrast-enhanced breast magnetic resonance imaging (MRI) in pregnancy remains controversial [115]. The distant staging includes X-ray test, liver ultrasound, and non-contrast supine MRI to check for lung, liver, and bone metastasis, respectively [114,116]. Based on the imaging results, biopsy or core needle biopsy defines the diagnosis of a breast mass [117]. After diagnosis, treatment can be given postpartum if the patient is in near term; otherwise, treatment should not be postponed until after the termination of pregnancy [118]. Breast cancer surgery may be performed safely during any stage of pregnancy with sentinel lymph node staging, favoring breast-conserving surgery when feasible, while radiotherapy may start after delivery [119]. Regarding systemic therapy, in PABC, the same guidelines as in non-pregnant patients should be followed. Chemotherapy, including anthracyclines, fluoropyrimidines, taxanes, and platinum agents, is feasible after the first trimester, and it can be carried out in both neoadjuvant and adjuvant settings, taking into account the gestational age and overall treatment plan, including need of HER2 targeted therapy (contraindicated in pregnancy), and indication for radiotherapy [119]. Regarding the possible reproductive ICIs toxicity, the PD-1/PD-L1 axis is critical in maintaining maternal fetal immunotolerance. Preclinical studies and limited clinical experience (case reports) have shown adverse effects on the fetus from the use of anti PD-1/anti PD-L1 antibodies. Therefore, the anti PD-1/anti PD-L1 agents have been categorized as category D in relation to the risk on pregnancy by the FDA, reporting that potential benefits may warrant use of these drugs in pregnant patients despite potential risks [120]. 

Another issue is the impact of chemoimmunotherapy on fertility in young patients [121]. The risk of chemotherapy-induced menopause and infertility is influenced by both the type of chemotherapy and the age of the patient [122]. In patients below 40 years of age, the risk of permanent chemotherapy-induced ovarian function failure after an anthracycline and taxane-based chemotherapy regimen is intermediate is about 40–60% [123]. Immune checkpoint inhibitors have an uncertain impact on fertility. These drugs may act either directly or indirectly on the reproductive organs [120], inducing primary or secondary hypogonadism, the latter often with panhypopituitarism (with hypothyroidism and secondary adrenal insufficiency) [124]. Furthermore, secondary hypogonadism may arise without panhypopituitarism, i.e., without alteration of other endocrine axes [125]. Of note, the frequency of hypogonadism, which does not appear to be a common adverse effect, may be underestimated due to the lack of routine testing of sexual hormones [125,126]. Given the paucity of data on the impact of immunotherapy on fertility, it is unclear whether fertility preservation strategies adopted for patients treated with chemotherapy, such as concomitant administration of ovarian suppression, or cryopreservation of embryos/oocytes, are also indicated in these cases [120]. 

In summary, treatment with immunotherapy in pregnancy requires a very accurate multidisciplinary discussion, taking into account potential benefits on oncological outcome but also the non-negligible impact on gestation, with a potential risk of congenital anomalies, abortion, and premature birth [120]. Regarding the impact of immunotherapy on fertility, more information on the gonad toxicity of ICIs needs to be acquired through the routine use of dedicated biochemical tests.

## 7. Conclusions

Compared with other breast cancer subtypes, TNBC is characterized by worse prognosis and early relapses. Thus, intensive treatment strategies have been developed so far for early stage disease. Unfortunately, the systemic treatment of TNBC has solely relied on chemotherapy for decades. However, new biologic and targeted agents are increasingly emerging, reshaping the treatment algorithms for this disease also in the early setting. 

Carboplatin added to anthracycline/taxane NACT has shown significant improvements in pCR and, recently, long-term outcomes, at the cost of higher myelotoxicity. 

Immunotherapy with ICI further improved pCR rates and long-term outcomes, especially in high risk disease, regardless of PD-L1 status. The discordance in some results of these trials could mainly derive from the chemotherapy backbone adopted in NACT, since anthracyclines more than others would be able to induce immune tumor death, activating the immune system against the tumor. Results from ongoing trials are awaited to better define the best chemotherapy backbone and ICI duration, especially after surgery, where the true impact on survival outcomes for adjuvant immunotherapy has to be better elucidated. Given the high costs and potential immune-mediated toxicities with this therapeutic strategy, the discovery of biomarkers for a more adequate patient selection is of the upmost importance. Subgroup analyses of the randomized trials show a greater absolute benefit with the use of ICIs compared to NACT alone for patients with more advanced disease stages, and especially with lymph node metastatic involvement. The absolute gain in pCR rate resulting from the ICI addition to NACT increases from 6–9% for node-negative patients, to 20–26% for node-positive patients. A possible explanation lies in the enhanced immunosuppression of the TME when TDNs are involved by cancer cells, which could be better eradicated by ICIs use.

The role of other agents in the neoadjuvant setting, such as PARP inhibitors, is yet to be defined.

Finally, novel antibody–drug conjugates such as sacituzumab govitecan have shown impressive results in metastatic disease, and further research is ongoing or warranted to elucidate their role in early-stage TNBC. 

## Figures and Tables

**Table 1 cancers-14-04064-t001:** Neoadjuvant randomized trials exploring the addition of immune checkpoint inhibitors in triple-negative breast cancer.

Study	Phase	Patients	Neoadjuvant Arms	Adjuvant Phase	Primary Endpoint	pCR Rate	pCR Difference Rate	*p*-Value	Survival Results
GeparNuevo [58,59] [Loibl, Ann Oncol 2019; Loibl, JCO, 2021]	II	88 86	Durvalumab + Nab-P → Durvalumab + ddEC Nab-P → ddEC	Physician’s choice	pCR (ypT0 ypN0)	53.4% 44.2%	+9.2%	0.224	HR 0.54 (*) (95% CI, 0.27 to 1.09)
NeoTRIPaPDL1 [60] [Gianni, Ann Oncol 2022]	III	138 142	Atezolizumab + Nab-P + Cp Nab-P + Cp	Anthracyclines (**)	EFS (***)	43.5% 40.8%	+2.7%	0.66	NR
KEYNOTE-522 [61,62] [Schmid, NEJM 2020; Schmid, NEJM 2022]	III	401 (****) 201 (****)	Pembrolizumab + P + Cp → Pembrolizumab + AC or EC P + Cp → AC or EC	Pembrolizumab/placebo for up to 9 cycles (capecitabine not allowed in either arm)	pCR (ypT0/Tis ypN0) EFS	64.8% 51.2%	+13.6%	< 0.001	HR 0.63 (*****) (95% CI, 0.48 to 0.82)
IMpassion031 [63] [Mittendorf, Lancet 2020]	III	165 168	Atezolizumab + Nab-P → Atezolizumab + ddAC Nab-P → ddAC	Atezolizumab/placebo for up to 11 cycles (capecitabine allowed in both arms)	pCR (ypT0/Tis ypN0)	58% 41%	+17%	0.0044	HR 0.76 (******) (95% CI, 0.40 to 1.44)

pCR, pathological complete response; EFS: event-free survival; Nab-P, nab-paclitaxel; dd, dose dense; EC, epirubicin and cyclophosphamide; NR, not reported; Cp, carboplatin; AC, doxorubicin and cyclophosphamide; P, paclitaxel; HR, hazard ratio; CI, confidence interval. * Invasive disease-free survival result; *p*-value 0.0559. ** In the adjuvant phase of the NeoTRIPaPDL1, the administration of anthracycline regimens was at investigator’s choice. *** In the NeoTRIPaPDL1, the pCR rate (ypT0/Tis ypN0) was a secondary endpoint. **** In the KEYNOTE-522, 1174 patients were randomized; the first interim pCR analysis was conducted on the first 602 patients who underwent randomization. ***** EFS result, *p*-value *<* 0.001. ****** EFS results are not mature and refer to a median follow-up of 20.6 months (HR 0.76; 95% CI, 0.40 to 1.44).

**Table 2 cancers-14-04064-t002:** Pathological complete response rate based on the PD-L1 status (positive vs. negative) in neoadjuvant immunotherapy randomized trials.

Study	Trial Arms	pCR Rate in PD-L1+	pCR Difference Rate	pCR Rate in PD-L1-	pCR Difference Rate
GeparNuevo [58] [Loibl, Ann Oncol 2019]	Durvalumab + Nab-P → Durvalumab + ddEC Nab-P → ddEC	58% 50.7%	+7.3%	44.4% 18.2%	+26.2%
NeoTRIPaPDL1 [60] [Gianni, Ann Oncol 2022]	Atezolizumab + Nab-P + Cp Nab-P + Cp	87%(IC2/3); 56.2% (IC1) (*) 72%(IC2/3); 44% (IC1) (*)	+15% (IC2/3) +12.2% (IC1)	35.1% (IC0) 41.1% (IC0)	−6%
KEYNOTE-522 [61] [Schmid, NEJM 2020]	Pembrolizumab + P + Cp → Pembrolizumab + AC or EC P + Cp → AC or EC	68.9% 54.9%	+14%	45.3% 30.3%	+15%
IMpassion031 [63] [Mittendorf, Lancet 2020]	Atezolizumab + Nab-P → Atezolizumab + ddAC Nab-P → ddAC	69% 49%	+20%	48% 34%	+14%

pCR, pathological complete response; PD-L1+, PD-L1-positive; PD-L1-, PD-L1-negative; Nab-P, nab-paclitaxel; dd, dose dense; EC, epirubicin and cyclophosphamide; Cp, carboplatin; IC, immune cells; AC, doxorubicin and cyclophosphamide; P, paclitaxel. * In the NeoTRIPaPDL1, the pCR rates were analyzed according to PD-L1 status defined as IC0 (PD-L1 expression in < 1% of tumor infiltrating immune cells) or IC1/2/3 (PD-L1 expression in ≥ 1% of tumor infiltrating immune cells).

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
