# Peer review of "Updated Neoadjuvant Treatment Landscape for Early Triple Negative Breast Cancer: Immunotherapy, Potential Predictive Biomarkers, and Novel Agents"

_cancers, 2022, doi:10.3390/cancers14174064_

Round 1

Reviewer 1 Report

This article focuses on an important topic related to the clinical and therapeutic implications of neoadjuvant therapy in TNBC. 

Since there is little data in the literature (only five main RCT) and considering the authors' experience in this field, this study is essential in identifying the mechanisms that will underlie the best experimental and clinical practices. Overall, this manuscript is well written and documented. Therefore, I recommend minor revisions. 

However, some suggestions could improve the quality of the article:

-       Line 47 without the extra symbol 1

-       Line 551 Please specify avelumab in A-BRAVE trial and pembrolizumab in SWOG S1418/BR006

-       Please specify NCT04434040 – ASPRIA trial, NCT04595565 - SASCIA trial

-       A suggestion would be to introduce a paragraph related to pregnancy and TNBC, as well as how chemo-immunotherapy influences fertility.

-       Another trial is c-TRAK TN - a phase II, multicentre, randomized trial with ctDNA screening.

-       A bibliographic reference that can be inserted would be the following - Immunotherapy for early triple negative breast cancer: research agenda for the next decade

https://doi.org/10.1038/s41523-022-00386-1

Reviewer 2 Report

It is a well written paper covering TNBC treatment and options, however there are important and similar reviews in the same topic published and are missed (below). Also it would be useful to highlight how this review is different from up-to-date and published literature. 

Bianchini, G., De Angelis, C., Licata, L. et al. Treatment landscape of triple-negative breast cancer — expanded options, evolving needs. Nat Rev Clin Oncol 19, 91–113 (2022). https://doi.org/10.1038/s41571-021-00565-2

Bou Zerdan M, Ghorayeb T, Saliba F, Allam S, Bou Zerdan M, Yaghi M, Bilani N, Jaafar R, Nahleh Z. Triple Negative Breast Cancer: Updates on Classification and Treatment in 2021. Cancers (Basel). 2022 Feb 28;14(5):1253. doi: 10.3390/cancers14051253. PMID: 35267561; PMCID: PMC8909187.

Medina MA, Oza G, Sharma A, Arriaga LG, Hernández Hernández JM, Rotello VM, Ramirez JT. Triple-Negative Breast Cancer: A Review of Conventional and Advanced Therapeutic Strategies. Int J Environ Res Public Health. 2020 Mar 20;17(6):2078. doi: 10.3390/ijerph17062078. PMID: 32245065; PMCID: PMC7143295.
